# Effect of environmental factors in reducing the prevalence of schistosomiasis in schoolchildren: An analysis of three extensive national prevalence surveys in Brazil (1950–2018)

**Mariana Cristina Silva Santos**[1]*, **Guilherme Lopes de Oliveira**[2], **Sueli Aparecida Mingoti**[3], **Léo Heller**[1]

**1** Instituto René Rachou, Fiocruz Minas, Belo Horizonte, Minas Gerais, Brazil, **2** Departamento de Computação, Centro Federal de Educação Tecnológica de Minas Gerais, Brazil, **3** Departamento de Estatística, Universidade Federal de Minas Gerais, Brazil

☯ These authors contributed equally to this work.

* santos.marianacs@gmail.com

**Data Availability Statement:** All relevant data are within the manuscript and its Supporting Information files.

## Abstract

### Background

Over seven decades, Brazil has made admirable progress in controlling schistosomiasis, and a frequent question about the explanation for this reduction refers to the effect of improving environmental factors in the country. This article seeks to identify factors related to the change in the epidemiological situation of schistosomiasis mansoni infection by analyzing three national prevalence surveys conducted since 1950.

### Methodology/principal findings

This is an ecological study analyzing an unbalanced panel of data based on national surveys and considering the municipality as the unit of analysis. The sample consisted of 1,721 Brazilian municipalities, in which a total of 1,182,339 schoolchildren aged 7–14 were examined during the three periods corresponding to each survey (1947–1953, 1975–1979, and 2010–2015). The percentage of municipalities with zero cases of schistosomiasis was: 45.4%, 54.2% and 73.7%, respectively for those periods. A zero-inflated Poisson regression model, with fixed and random effects, was fitted to assess the association between candidate factors and disease prevalence using a significance level of 5%. There was a significant decrease in disease prevalence between the first and last periods analyzed (RR 0.214, CI 0.184–0.249), with a protective association with access to sanitation (RR 0.996, CI 0.994–0.998), urbanization (RR 0.991, CI 0.989–0.993), and living in own households (RR 0.986, CI 0.983–0.989); and an inverse association with piped water supply (RR 1.010, CI 1.008–1.011).

**Funding:** This work was carried out with the support of the Coordination for the Improvement of Higher Education Personnel – Brazil (CAPES) – Financing Code 001 (https://www.gov.br/capes/pt-br), which granted financial aid in the form of a scholarship granted to MCSS. The financier did not participate in the study design, data collection and analysis, publication decision or manuscript preparation.

**Competing interests:** The authors have declared that no competing interests exist.

## Conclusion

The findings of this study indicate a decrease in the prevalence of schistosomiasis over seven decades in schoolchildren from the analyzed Brazilian municipalities, associated with environmental factors and social conditions. The increased access to piped water in the municipalities apparently triggers other ways of contact with unsafe water bodies, generating new transmission routes and suggesting the need for a systemic approach concerning contact with water.

### Author summary

Schistosomiasis mansoni is a neglected tropical disease caused by infection from parasitic worms of the species *Schistosoma mansoni*. Due to the complexity of the mechanism of transmission and maintenance of schistosomiasis, several preventive actions on diverse conditioning factors can promote disease control. Active search, timely treatment of cases, stool tests, and epidemiological investigations are the initial actions under programs for epidemiological surveillance of the disease. Thus, national surveys on prevalence of the disease covering a large time span can provide valuable information about its epidemiological pattern over the years. Our study addressed three national surveys with historical coverage (1947–1953, 1975–1979, and 2010–2015) that mapped the prevalence of the disease in children aged 7–14 for nearly seven decades. We also employed statistical models to investigate which environmental, economic, or demographic factors are associated with the disease at municipal level. The results showed that the decrease in schistosomiasis from the 1950s to the 2010s was statistically significant and suggests that improvements in water supply and sanitation require structured and systemic approaches for controlling the transmission of schistosomiasis.

## Introduction

Over the last decades, several countries have tried to control neglected tropical diseases, including schistosomiasis, by establishing measures to intensify their management. Schistosomiasis is endemic in at least 52 countries [1], affecting approximately 240 million people worldwide. This disease is endemic in ten countries on the American continent. However, only Brazil and Venezuela needed to apply preventive chemotherapy for their population in 2020, including more than 2.2 million school-age children [2]. In addition to chemotherapy, which is not sufficient and accessible to all, the World Health Organization (WHO) recommends several strategies to control and eliminate the disease. These measures include access to safe drinking water, improvements in sanitation, health education, and hygiene, besides environmental and disease control management, even though considering that WaSH interventions (water, sanitation, and hygiene) are expected to provide modest benefits in limiting *Schistosoma* transmission" [3].

Prevalent in tropical and subtropical areas, especially in poor communities without access to drinking water and adequate sanitation, the disease caused by trematode helminths of the genus *Schistosoma* has epidemiological importance. The epidemiology of the disease is especially relevant in children since the absence of infection in this age group would mean the possible interruption of the transmission. On the other hand, eliminating the disease from the

population, including adults, especially workers living in large endemic areas, requires improved household and environmental conditions. Among them, access to safe and continuous water and improved sanitary facilities that allow for better conditions dwelling can have an important role in breaking the disease cycle, interrupting the release of eggs in the environment and avoiding access to surface water for water supply [4].

In Brazil, the epidemiology of *Schistosoma mansoni* infection shows that social and, environmental conditions, drug treatment and access to health service contribute to a reduction in the prevalence rate [5]. Although the relationship between schistosomiasis infection and sanitary conditions has been showed in local or regional scales, nationwide and longitudinal studies can contribute to understanding disease dissemination as well as its explanatory factors throughout the Brazilian territory. Approaches to exploring and understanding the role of environmental, biological and medical interventions, as well as historical, socioeconomic, and cultural determinants, crucial for assessing this complex disease [6].

Brazil has an extensive experience in conducting surveys on the prevalence of schistosomiasis, covering a wide range of the country and an extended time of approximately seven decades. The first of these surveys was carried out in the 1950s [7,8]. Given the epidemiological and social impact of schistosomiasis on the population, other two national surveys were conducted: by the Special Schistosomiasis Control Program (PECE) (*Programa Especial de Controle da Esquistosomose*)in the 1970s [9] and the National Survey on the Prevalence of Schistosomiasis and Soil-Transmitted Helminth Infections (INPEG) (*Inquérito Nacional de Esquistosomose e Geo-helmintose*) in the 2010s. Throughout these seven decades, a reduction in prevalence could be observed [10]. However, these data demonstrate that schistosomiasis is still epidemiologically relevant [11] since, from the point of view of the infected patient and public health, there should be no acceptable level of morbidity due to this disease [3].

Hence, this study aimed to analyze the behavior of the prevalence of schistosomiasis and the impact on prevalence of access to water and sanitation services. The analysis is based on those three surveys conducted in Brazilian municipalities over seven decades.

## Methods

### Ethics statement

The current study used data from three national survey on prevalence schistosomiasis in schoolchildren. These data are anonymous and available for research purposes by the Brazilian government. Moreover, this study was conducted exclusively with secondary and aggregated data, publicly accessible and in accordance with resolutions of the National Health Council No. 466/2012 [12] and No. 510/2016 [13], exempt from evaluation by the Research Ethics Committee.

### Study design

The epidemiological design of the research consists of a prospective study, covering three periods with observational ecological data. The outcome variable was the municipal prevalence of schistosomiasis in schoolchildren from seven to 14 years old.

### Studied period and data source

Data were extracted from the three Brazilian surveys of schistosomiasis prevalence, as follows:

i. The National Helminthological Survey of Schoolchildren (IHE) (*Inquérito Helmintológico Escolar*) by Pellon & Teixeira, conducted from 1947–1953 in two phases [6,7]. The first phase included 11 states considered endemic for the disease, with a sampling plan that

addressed locations of more than 1,500 inhabitants in which 440,786 schoolchildren were examined. In the second phase, locations of more than 1,250 inhabitants of five non-endemic states were included, and 174,192 schoolchildren were examined. In both phases, all regions of the country were sampled, except for the North region. In this way, 1,190 locations were surveyed, totaling 614,978 students examined.

ii. Survey by the Special Schistosomiasis Control Program (PECE) (*Programa Especial de Controle da Esquistosomose*) conducted from 1975–1979 [9]. This survey consisted of a non-probabilistic sample of 327 municipalities in 18 states and areas that were disease-free or endemic, in which 447,779 schoolchildren aged 7–14 were examined. This survey took place in municipalities where the program had been implemented by the Ministry of Health and included all municipalities that adhered to PECE. The criteria for inclusion of schools and students were based on the decennial census and an active search in school classes [14,15].

iii. INPEG conducted from 2010–2015 [10]. This survey also considered schoolchildren aged 7–14 by applying a cluster sampling plan, with areas categorized in three endemic levels (municipalities in non-endemic, low prevalence, and high prevalence areas) and four categories of population size (fewer than 20,000, between 20,000 and 150,000, between 150,000 and 500,000, and more than 500,000 inhabitants). Thus, samples were drawn from those stratums to determine the analyzed municipalities, elementary schools, and school classes. As a result, 521 municipalities representing all Brazilian states were analyzed. The amount of tests in each municipality ranged from 60% to 100% and in nine states it was higher than planned. In total, 197,564 schoolchildren aged 7–14 were examined.

The broad extension of the Brazilian territory affected the implementation time of each of the three surveys. Therefore, the impossibility of collecting data in just one year led to the need for around five years of data gathering for each survey. Fig 1 describes the surveys, including their respective sampling strategies. S1 Note provides additional details on characteristics of each survey and their specific features.

For obtaining intercensal estimates, data related to the explanatory variables were collected from the 1950, 1960, 1970, 1980, 2000, and 2010 demographic censuses of the Brazilian Institute of Geography and Statistics (Instituto Brasileiro de Geografia e Estatística–IBGE) and from the Institute of Applied Economic Research (*Instituto de Pesquisa Econômica Aplicada*–IPEADATA) (Table 1).

## Inclusion and exclusion criteria

The intense evolution of the political-administrative organization of Brazilian states and municipalities over the decades, reflected in the number of currently existing municipalities, led to the adoption of inclusion and exclusion criteria for this study defined as: (i) sampled municipalities with territorial delimitation compatible with the demographic censuses of each analyzed period; (ii) sampled municipalities and/or municipal districts that, even incorporated to or emancipated from other municipalities or districts during the period of the three surveys (1947–1953, 1975–1979 and 2010–2015), had available legislative and historical information on their establishment or division process; (iii) sampled municipalities that met the criterion of quality of registration. Subsequently, according to the assumed criteria, the municipalities in which it was not possible to detail the evolution of their establishment, fusion, or incorporation, as well as those lacking enough records for the explanatory variables, were excluded. Fig 1 and S2 Note show the complete description of the methodological inclusion and exclusion criteria.

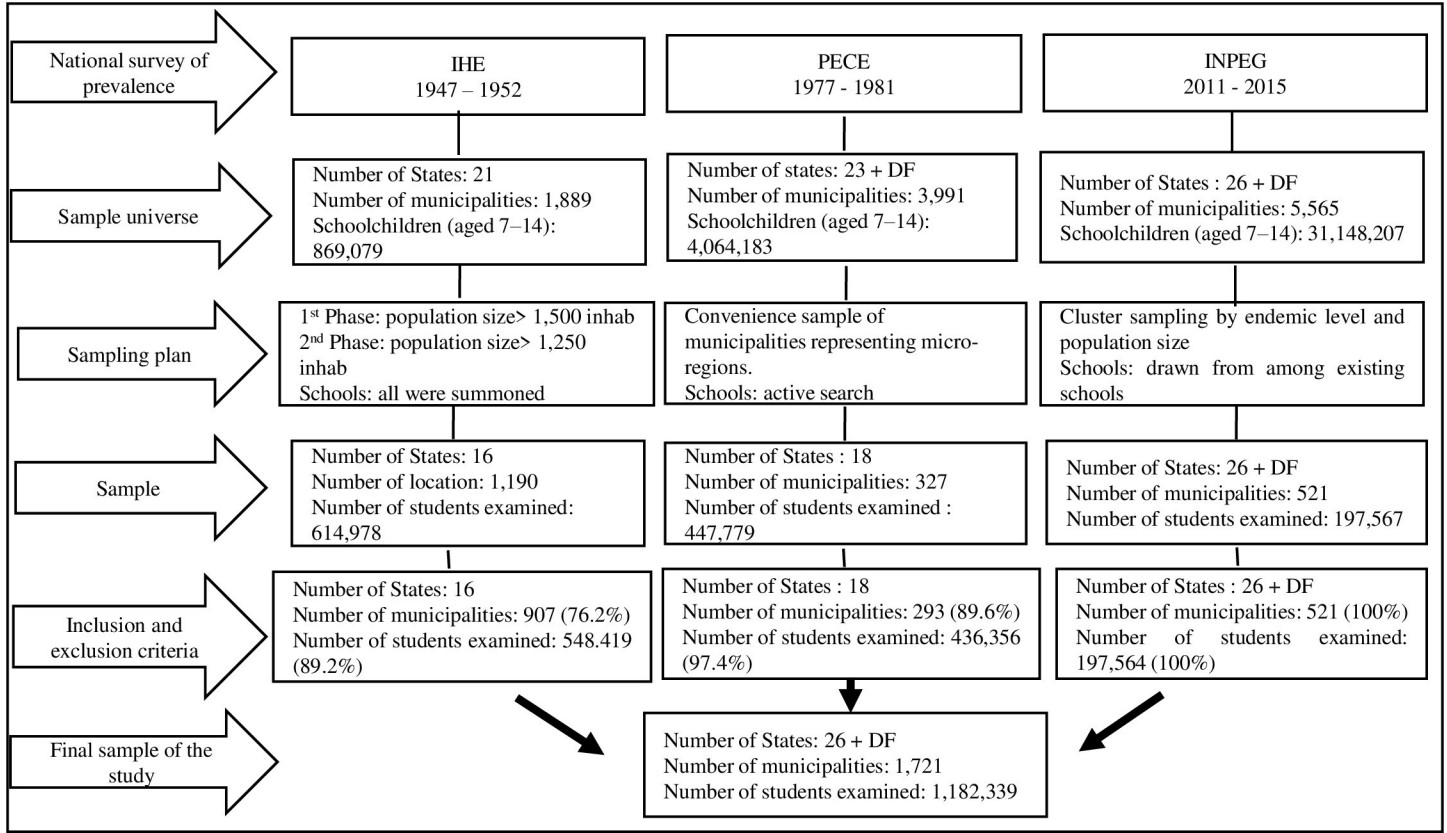

**Fig 1. Descriptive flowchart of the three national surveys on the prevalence of schistosomiasis mansoni in Brazil.** IHE: National Helminthological Survey of Schoolchildren. PECE: Special Schistosomiasis Control Program. INPEG: National Survey of Prevalence of Schistosomiasis and Soil-transmitted helminth infections. DF: Federal District.

## Outcome variable

The outcome variable of the study was the prevalence of infection with *Schistosoma mansoni* in samples of schoolchildren from 7 to 14 years old per municipality (Table 1).

## Independent variables

The independent (explanatory) variables consisted of coverage of water supply and sewerage, and municipalities sociodemographic and socioeconomic variables such as population size, percentages of urbanization and literacy, *per capita* gross domestic product, and the survey period. These variables were defined considering the factors related to infection as indicated in the literature and the context and availability of data in the information systems for each period. Relevant factors related to the disease, such as family income, coverage of deworming treatment, water treatment for inactivating human schistosome cercariae or chemical mollus-cicide treatment, malacological surveys, and family or school hygiene practices, could not be included since there are not enough available data from all studied municipalities in the different periods, mainly in the 1950s and 1970s.

The reference year adopted for each of the survey periods were 1950, 1977, and 2013 to facilitate notation. Projection and/or interpolation techniques were used in cases where information about the explanatory variables was not available in the reference year. For 1950, we applied a projection by the population trend method—*AiBi* projection or Apportionment

**Table 1. Description of evaluated outcome and explanatory variables, periods, and data source.**

| Variable | Description | Source | Period | | |
|---|---|---|---|---|---|
| | | | IHE | PECE | INPEG |
| Prevalence of schistosomiasis | Number of students with positive stool tests / Total number of students aged 7–14 examined | National surveys | 1947–1953[a] | 1975–1979[a] | 2010–2015[a] |
| % of water supply | Number of dwellings with internal piped water supply from the general distribution network / Total number of dwellings | IBGE Census | 1950 | 1970 and 1980[b] | 2000–2010[c] |
| % of sanitary sewerage | Number of dwellings with sanitary facilities with drainage connected to the general sewage networks/ Total number of dwellings | IBGE Census | 1950 and 1960[d] | 1970–1980[b] | 2000–2010[c] |
| % of urbanization | Number of inhabitants in the urban area / Total number of inhabitants | IBGE Census | 1950 | 1970 and 1980[b] | 2000–2010[c] |
| % literacy rate | Number of literate people aged 15 years old or older / Total population of the same age group | IBGE Census | 1950 | 1970–1980[b] | 2000–2010[c] |
| % Occupancy condition of households | Number of permanent households in occupancy and owning conditions / Total number of permanent households | IBGE Census | 1950 | 1970–1980[b] | 2000–2010[c] |
| Municipal GDP per capita | Municipal GDP at constant prices–R$ 1,000 per year 2000s/ total population in the municipality | IPEADATA | 1949 and 1959[b] | 1970–1975–1980[b] | 1999–2010[c] |
| Period | Variable with three categories corresponding to the periods of the surveys | – | 1950 (reference) | 1977 | 2013 |

[a:]Year interval used to define 1950, 1977, and 2013 midpoints for the collection and treatment of explanatory variables.

[b:] explanatory variables calculated by applying linear interpolation techniques.

[c:] explanatory variables calculated using linear and polynomial interpolation and extrapolation techniques.

[d:]values calculated by the population trend method or Apportionment Method (*AiBi* projection) [15]. National Helminthological Survey of Schoolchildren *(Inquérito Helmintólogico Escolar*–IHE). National Survey on the Prevalence of Schistosomiasis and Soil-transmitted helminth infections (INPEG). Brazilian Institute of Geography and Statistics (IBGE). Institute for Applied Economic Research (IPEADATA). Ministry of Health (MS). Special Schistosomiasis Control Program (PECE). Gross Domestic Product (GDP)

Method—for the sanitary sewage [16,17]; and a projection by interpolation for the municipal gross domestic product (GDP) *per capita* using data from 1949 and 1959. For explanatory variables in the 1977 period, estimates were made using linear interpolation techniques. For 2013, interpolation estimates were performed for 2000 and 2010, and then we extrapolated linear and geometric growth for 2011, 2012, and 2013. The projections for the years 1977 and 2013 were adopted because the Brazilian census information is collected every decade, therefore, in a non-annual series [18]. S2 Note provides additional details about the techniques used for each variable, explanations of the use of the *AiBi* technique and the number of observations involved.

## Data analysis

As the national surveys were carried out in different periods and using different strategies, the sampled municipalities were not the same for all periods, resulting in an unbalanced data panel with different municipalities in each sampling period. Based on that, we conducted a prospective study covering three periods with observational ecological data to evaluate trends in the prevalence rates of the infection over time and their associations with economic, health, and social indicators for a total of 1,721 municipalities sampled during the periods represented by the reference years 1950, 1977, and 2013.

Descriptive analyses were performed for the municipal data in each period. In the inferential analyses, multilevel statistical models were fitted to estimate the prevalence of schistosomiasis considering data from the 1,721 sampled municipalities. According to the official territorial division proposed by the IBGE, the Brazilian political-administrative organization is

divided into five macro-regions, which include 26 federal units (states) plus the Federal District and 5,570 municipalities. In order to consider this hierarchical characteristic of the data, we applied Generalized Linear Mixed Models (GLMMs) with random effects related to three levels: regions (Level 1,with 5 categories South, Southeast, North, Northeast and Midwest); states (Level 2, representing the 26 federative units plus the Federal District); and municipalities (Level 3, related to the 1,398 different municipalities included in the study with observation in at least one of the three surveys). These three hierarchical levels of data were incorporated into the random intercepts of the GLMMs to allow the joint modeling of data the three periods.

The GLMMs also included fixed effects related to the independent variables previously described. We considered the Poisson and Negative Binomial distributions in the analyses, with and without zero inflation for both cases [19–22]. Thus, we allowed the modeling of different data characteristics, such as over dispersion and zero inflation. Detailed discussion on models definition is provided in S3 Note. The count of the number of positive cases in each municipality was considered the outcome variable and the total number of examined students was used as an offset, responsible for controlling the number of cases per municipality. Logistic regression was used to adjust for the excess of zero.

A backward selection procedure was used to identify the significant fixed effects, considering a 25% significance level for the removal of an explanatory variable. Thus, at each step of the analysis, the explanatory variable with the highest p-value, among those with a p-value>0.25, was removed from the model. After such a procedure, no further variable selection was necessary, as all variables retained were significant at the 5% significance level. The final regression model (Poisson or Negative Binomial distribution with or without zero inflation) was chosen according to the following criteria: (a) lower residual variance; (b) lower values of Akaike (AIC) and Bayesian (BIC) Information Criteria [23].

The software EPI INFO 7.1.1 and Microsoft Office Excel 2010 were used for database construction. Descriptive and inferential analyses were performed in the software *R* using the statistical package *glmmTMB* [24].

## Results

### Descriptive analysis

Table 2 shows the results of the descriptive analysis for the prevalence of schistosomiasis. Despite the large range, the mean prevalence of infection decreased between the three analyzed periods, with 8.3% for reference period 1950 (SD 17.2), 4.8% for 1977 (SD 12.4), and 0.8% for 2013 (SD 3.5). In addition, the median and amplitude of prevalence were 0.2 and 90.9 in 1950; 0.0 and 71.2 in 1977; and 0.0 and 50.0 in 2010–2015. The percentage of municipalities with zero cases of schistosomiasis were 45.4% for 1950, 54.2% for 1977 and 73.7% for 2015.

Table 2 also shows descriptive statistics for the explanatory variables that composed the study. They all showed remarkable increasing values between 1947–1953 and 2010–2015, especially the sanitary variables related to water supply and sewerage coverages. On average, urbanization varied from 25.6% to 68.4% (a 2.6-fold increase); literacy from 38.6% to 84.1% (2.1-fold increase); coverage of water supply network from 6.5% to 71.6% (an 11-fold increase); coverage of sewerage from 2.6% to 31.0% (an 11.9-fold increase); condition of occupancy conditions of households from 54.9% to 76.3% (a 1.4-fold increase); and GDP from 0.91 to 5.77 (BRL) (a 6.3-fold increase). Additionally, for the 41 municipalities common to the three surveys, the percentage decrease in prevalence between the 1947–1953 survey and the 2010–2015 survey ranged from 0.1 percentage point (p.p) to 77.4 p.p., with only three municipalities presenting small positive percentage difference between 0.4 and 0.1 p.p.

**Table 2. Descriptive statistics on the prevalence of schistosomiasis per 100 students and independent variables per study period in the 1,721 sampled Brazilian municipalities.**

|  | 1947–1953 (n = 907) | | | | 1975–1979 (n = 293) | | | | 2010–2015 (n = 521) | | | |
|---|---|---|---|---|---|---|---|---|---|---|---|---|
| **Dependent variable** | Mean | SD | Median | Range | Mean | SD | Median | Range | Mean | SD | Median | Range |
| Prevalence of schistosomiasis | 8.3 | 17.2 | 0.2 | 90.9 | 4.8 | 12.4 | 0.0 | 71.2 | 0.8 | 3.5 | 0.0 | 50.0 |
| **Independent variables** | Mean | SD | Median | Range | Mean | SD | Median | Range | Mean | SD | Median | Range |
| %Urbanization | 25.6 | 17.4 | 20.6 | 97.0 | 47.4 | 24.5 | 41.6 | 96.6 | 68.4 | 23.2 | 69.1 | 86.3 |
| %Literacy | 38.6 | 15.5 | 36.8 | 77.7 | 59.8 | 17.3 | 60.6 | 76.3 | 84.1 | 10.0 | 85.7 | 88.3 |
| %Water supply | 6.5 | 10.4 | 1.5 | 73.0 | 30.0 | 22.0 | 24.9 | 90.7 | 71.6 | 21.4 | 75.2 | 100.0 |
| %Sewerage | 2.6 | 4.7 | 0.0 | 28.8 | 8.6 | 16.2 | 0.0 | 73.1 | 30.6 | 30.8 | 20.5 | 98.7 |
| % Occupancy condition of the households | 54.9 | 21.1 | 55.4 | 91.5 | 66.2 | 15.1 | 66.3 | 85.4 | 76.3 | 9.1 | 76.7 | 51.9 |
| Municipal GDP per capita | 0.9 | 0.7 | 0.7 | 6.0 | 2.9 | 2.4 | 2.2 | 13.6 | 5.8 | 5.6 | 4.2 | 49.2 |

Range: difference between maximum and minimum values. SD: Standard Deviation. GDP: Gross Domestic Product, in 1,000 Brazilian Reais (BRL), adjusted to the base year of 2000. n = number of sampled municipalities.

Table 3 shows the hierarchical (multilevel) description adopted in the study, detailing the distribution of the number of municipalities according to regions and federative units state in each analyzed period. Regarding the distribution of the studied municipalities along the five geographical regions of the country, 758 (44.0%) are from to the Northeast, 506 (29.4%) from the Southeast, 206 (11.9%) from the South, 153 (8.9%) from the Midwest, and 98 (5.7%) from the North region. The Northeast region had the highest percentages of municipalities in each survey, following by the Southeast. In 1947–1953 (n = 907), the survey included 418 (46.0%) municipalities from the Northeast, 317 (35.0%) from the Southeast, 103 (11.4%) from the South, 69 (7.6%) from the Midwest, and no samples from the North region. For 1975–1979 (n = 293), 114 (38.9%) sampled municipalities belonged to the Northeast region, 73 (24.9%) to the Southeast, 50 (17.1%) to the South, 40 (13.7%) to the Midwest, and 16 (5.3%) to the North. Finally, in 2010–2015 (n = 521), 226 (43.4%) municipalities were in the Northeast, 116 (22.3%) in the Southeast, 82 (15.7%) in the North, 53 (10.2%) in the South, and 44 (8.4%) in the Midwest regions.

## Statistical models

Because of the larger amount of municipalities with zero cases of schistosomiasis (45.4% for 1947–1953 period; 54.2% for 1974–1979 period and 73.7% for 2010–2015 period), models with and without the adjustment for excess of zeros were employed in order to verify the robustness and consistency of the analyses.

The results between the goodness-of-fit measures for the adjusted models (Poisson and Negative Binomial with and without zero-inflation) can be verified in Table A in S3 Note. The Poisson models presented lower residual variance than Binomial Negative models, being the Poisson zero-inflated specification the model with the lowest AIC and BIC values. Table 4 shows the Rate Ratio (RR) estimates for schistosomiasis infection, and the respective 95% confidence intervals (CI) obtained from the zero-inflated Poisson multilevel regression model.

The explanatory variables that remained in the model of the prevalence of schistosomiasis were the natural logarithm of population size, %Urbanization, %Occupancy condition of the domicile, %Water supply, %Sewerage, and the categorical variable related to the survey period (the 1947–1953 period was used as a reference for the analysis). For the zero-inflation logistic regression, only variable %Urbanization showed statistical significance.

**Table 3. Hierarchical levels and the distribution of the 1,721 sampled Brazilian municipalities (Level 1) included in the study according to state (Level 2) and region (Level 3) for each period.**

| Level 3 | Level 2 | Level 1 | | | | | |
|---|---|---|---|---|---|---|---|
| Region | State | Municipalities | | | | | |
| | | IHE (1947–1953) | | PECE (1975–1979) | | INPEG (2010–2015) | |
| | | n | (%) | n | (%) | n | (%) |
| Northeast | Alagoas | 21 | 2.3 | 10 | 3.4 | 24 | 4.6 |
| | Bahia | 123 | 13.6 | NA | – | 47 | 9.0 |
| | Ceará | 60 | 6.6 | 22 | 7.5 | 21 | 4.0 |
| | Maranhão | 29 | 3.2 | 16 | 5.5 | 23 | 4.4 |
| | Paraíba | 36 | 4.0 | 17 | 5.8 | 21 | 4.0 |
| | Pernambuco | 61 | 6.7 | 13 | 4.4 | 29 | 5.6 |
| | Piauí | 16 | 1.8 | 12 | 4.1 | 19 | 3.7 |
| | Rio Grande do Norte | 41 | 4.5 | 13 | 4.4 | 20 | 3.8 |
| | Sergipe | 31 | 3.4 | 11 | 3.7 | 22 | 4.2 |
| | Subtotal | 418 | 46.0 | 114 | 38.9 | 226 | 43.4 |
| North | | NA | Subtotal | NA | – | 10 | 1.9 |
| | Amapá | NA | Acre | NA | – | 5 | 1.0 |
| | Amazonas | NA | – | NA | – | 15 | 2.9 |
| | Pará | NA | – | 16 | 5.5 | 19 | 3.7 |
| | Rondônia | NA | – | NA | – | 13 | 2.5 |
| | Roraima | NA | – | NA | – | 7 | 1.3 |
| | Tocantins | NA | – | NA | – | 13 | 2.5 |
| | Subtotal | 0 | 0 | 16 | 5.5 | 82 | 15.7 |
| Midwest | Distrito Federal | NA | – | NA | – | 1 | 0.2 |
| | Goiás | 49 | 5.4 | 25 | 8.5 | 18 | 3.5 |
| | Mato Grosso | 20 | 2.2 | 7 | 2.4 | 12 | 2.3 |
| | Mato Grosso do Sul | NA | – | 8 | 2.7 | 13 | 2.5 |
| | Subtotal | 69 | 7.6 | 40 | 13.6 | 44 | 8.4 |
| Southeast | Espírito Santo | 18 | 2.0 | 10 | 3.4 | 16 | 3.1 |
| | Minas Gerais | 250 | 27.6 | 52 | 17.7 | 56 | 10.8 |
| | Rio de Janeiro | 49 | 5.4 | 11 | 3.7 | 21 | 4.0 |
| | São Paulo | NA | | NA | – | 23 | 4.4 |
| | Subtotal | 317 | 35.0 | 73 | 24.9 | 116 | 22.3 |
| South | Paraná | 58 | 6.4 | 7 | 2.4 | 21 | 4.0 |
| | Rio Grande do Sul | NA | – | 25 | 8.5 | 14 | 2.7 |
| | Santa Catarina | 45 | 5.0 | 18 | 6.1 | 18 | 3.5 |
| | Subtotal | 103 | 11.4 | 50 | 17.1 | 53 | 10.2 |
| Total | | 907 | | 293 | | 521 | |

NA: not analyzed. School Helminthological Survey (IHE). National Survey on the Prevalence of Schistosomiasis and Soil-transmitted helminth infections (INPEG). Special Schistosomiasis Control Program (PECE).

A negative value in the estimate of the effect of a variable indicates that an increase in its value results in a decrease in the prevalence of the infection. This was the case for the variables natural logarithm of population size (-0.148; p-value <0.001), %Urbanization (-0.009; p-value <0.001), % Occupancy condition of the households (-0.014; p-value <0.001), and % Sewerage (-0.004; p-value 0.001) in the modeling of prevalence. Based on the associated RR, the increase of one unit in the numerical value of these variables causes a decrease of 13.8%, 0.9%, 1.4%, and 0.4% in the estimated mean for the prevalence, respectively. We highlight that one unit

**Table 4. Results from the zero-inflated Poisson multilevel regression model fitted to assess the prevalence of schistosomiasis mansoni in the sampled Brazilian schoolchildren.**

| Coefficient | Poisson regression | | | |
|---|---|---|---|---|
| | RR | (CI 95%) | Estimate | P-value |
| Model constant (intercept) | - | - | -5.488 | <0.001 |
| LN Population | 0.862 | (0.825–0.901) | -0.148 | <0.001 |
| %Urbanization | 0.991 | (0.989–0.993) | -0.009 | <0.001 |
| % Occupancy condition of the household | 0.986 | (0.983–0.989) | -0.014 | <0.001 |
| %Water supply | 1.010 | (1.008–1.011) | 0.010 | <0.001 |
| %Sewerage | 0.996 | (0.994–0.998) | -0.004 | <0.001 |
| Year: 1975–1979 | 1.352 | (1.256–1.454) | 0.301 | <0.001 |
| Year: 2010–2015 | 0.214 | (0.184–0.249) | -1.542 | <0.001 |
| Coefficient | Zero-inflation logistic regression | | | |
| | OR | (CI 95%) | Estimate | P-value |
| Model constant (intercept) | - | - | -8.647 | <0.001 |
| %Urbanization | 0.976 | (0.961–0.991) | -0.025 | 0.002 |

Residuals Variance: 4,283.5. AIC: 11,162.2. BIC: 11,233.1. Reference year: 1950. CI: Confidence interval. LN: natural logarithm. RR: Rate Ratio. OR: odds ratio. Standard deviation of random effects: Municipality 2.136; State 2.066; Region 1.706.

increase in the natural logarithm scale corresponds to an increase of approximately 2.718 times in the original variable scale. On the other hand, the results showed an inverse effect on the prevalence of infection for the water supply variable, with a positive value for its estimated effect (0.010; p-value <0.001; and RR corresponding to an increase of only 0,1% per one unit increase in the numerical value of the variable). Concerning the categorical variable representing the survey periods, in comparison with period 1947–1953 (taken as reference in the regression model) a positive regression effect was estimated for 1975–1979 (0.301; p-value <0.001) and a negative effect was estimated for 2013 (-1.542; p-value <0.001). Although this result seems to indicate an increase in prevalence from 1947–1953 to 1975–1979, contradicting the descriptive analysis shown in Table 2, it should be noted that the behavior of the other explanatory variables is quite different between those periods. In fact, an analysis of the municipal prevalence estimated by the model provided similar and consistent results with those observed in the data, corroborating the adequacy of the adjusted model (see Table B in S3 Note).

The use of GLMMS allowed the joint modeling of data from all municipalities in the three sampling periods. In order to evaluate the robustness of this approach, we performed a sensitivity analysis involving the data subset composed of the 41 common municipalities between the three sampling periods (see Table C in S3 Note). The zero-inflated Poisson was the best fitted model and composed of the same explanatory variables to explain the prevalence of schistosomiasis as for the multilevel zero-inflated Poisson model presented in Table 4. The estimates of the coefficients and Rate Ratio (RR) are similar. We also performed a statistical analysis comparing the distribution of the municipalities of the three surveys according to the endemicity classification used in the sampling procedure of the third survey (2010–2015). The results indicated that, although there are differences in the form of data collection regarding the selection of municipalities, the three samples are comparable in terms of the endemicity degree criterion used in the 2010–2015 survey.

## Discussion

The analysis identified significant effects of environmental, economic, and demographic factors on the prevalence of schistosomiasis by evaluating its trend during the three national surveys. Hence, this study found significant associations between environmental factors and schistosomiasis. The descriptive analysis among the municipalities common to the three surveys indicated a decrease in the prevalence percentages for most of the analyzed municipalities (92.7%), when compared 1947–1953 and 2010–2015. The fitted statistical model also predicted a decreasing behavior in the prevalence among the three sampling surveys.

The results of the statistical model of this study showed that the environmental variables contributed significantly to the prevalence of schistosomiasis. The protective association between the expansion of sewerage coverage and the reduction of prevalence has been portrayed in epidemiological studies since the 1960s [25]. For instance, a significant association with the disease prevalence was found in households with any type of sewage disposal when compared to those using a safe sewage network (OR 1.8; CI 1.3–2.4) [26]. This result is in line with national and international studies, showing that improvement of sanitation was significantly associated with a decreased probability of infection [27,28]. Even when latrines were available, families' preference for their use also reduced the occurrence of the disease [29], which was found when households lacked a functional toilet [30,31].

Therefore, although Brazil had sanitary sewage networks in only 60.3% of its municipalities in 2017 [32], the impact of this service in the interruption of the disease is evident, as a sanitary barrier to fecal contamination in water bodies containing intermediate hosts. The results of this study validate the importance of public policies promoting the implementation of sanitation solutions. According to these results, if municipalities with a coverage of 20% of the sewage system, a common situation in some areas of the country, reach 100% coverage, a 27.4% (value obtained from the equation: $\exp(-0.004*80) = 0,726$) reduction in the average prevalence of schistosomiasis can be expected, which is an important outcome in terms of public health.

In addition, the treatment and supply of safe drinking water have been considered another environmental variable as an effective and lasting measure to prevent disease [26,33–35]. Some studies, in convergence with this research, found no significant association between drinking water supply and reduced prevalence of schistosomiasis [36–39]. Although schistosomiasis is not a waterborne disease, adequate water supply is expected to be positively associated with its control, by avoiding the need for individuals to have contact with surface water in order to fetch water for household supply. Thus, it is reasonable to assume that the presence of piped water should not pose a risk of transmission. However, although the results of this study indicate a controversial finding, three possible explanations can be put forward.

Firstly, the infrastructure for piped water supply has expanded over the decades, but this expansion has not guaranteed uninterrupted supply, or the quality of water supplied. Even in a more recent period, in 2006, the irregularity in water supply from the public network can affect about 80% of Brazilian municipalities in certain regions, like the state of Bahia, where schistosomiasis is endemic [40]. Moreover, the Northeast and Southeast regions, which presented the highest prevalence of the disease, exhibited the highest frequencies of systematic interruptions in the water supply in 2020, reaching 66.1% and 46.5%, respectively [41]. This intermittence can lead users to depend on contact with unsafe water sources, contacts that may even increase at day times of high schistosomiasis transmission. Consequently, even in municipalities with households supplied with piped water, there could be a high probability of infection by the disease. Intermittent water supply can disrupt family dynamics, a situation directly related to obligations that often still fall on women. In a society and economy marked by the sexual division of labor, this dynamic leads to the penalization mainly of women and their children, who end

up accompanying their mothers [42] in using unsafe water sources, a risk factor in the dynamics of schistosomiasis transmission, reported since the 1980s [43].

In addition, discontinuity of water supply produces other adverse effects, such as disruption of water networks designed for continuous supply, leading to leaks and deterioration of the water quality. Consequently, users adapt to meet adversities, highlighting the inequality and vulnerability to shortages to which a city or region is exposed [44]. An intervention study showed that the positive impact of piped water occurred only when the amount of water available was higher than 1,000 liters per person per year, i.e., the use of unsafe water can continue if only a small amount of water is provided or if there are interruptions due to precarious distribution systems [45].

The second explanation regarding disease prevalence despite the availability of piped water is related to a supply insufficiency for some households to eliminate other contact forms with surface water for domestic, leisure, behavioral, or labor use, such as fishing and irrigation. Eventually, the presence of piped water supply may free up more time for residents to perform these activities more frequently, increasing the risk of contamination. When disassociated from facilities for other home uses, such as laundry, sink, and shower, piped water supply can contribute to the continuity or increase of the behavior of accessing transmission sites [46,47]. Another aspect related to the water contact practices was demonstrated by a spatial community study verifying that the public water supply could potentially decrease dependence on surface water. However, this relationship was modified by the quality of the water from the sources of public supply, which was considered poor by domestic users [48,49].

Thirdly, an aspect probably not strongly related to our results although worthy of analysis, is the effect of the technology used in surface uptake, adduction, and water treatment on dermal contact and survival of infectious forms of the schistosome. Filtration and chlorination are widely used methods for water treatment in conventional and simplified treatment plants all over the country [50,51]. These processes are credited as likely to produce waters free of contamination from cercariae, depending on storage time, exposure temperature, chlorine concentration, or filtration rates, besides the concentration of cercariae itself [51]. However, there are no current guidelines for the specific care related to water treatment and its respective technical and operational infrastructure in endemic schistosomiasis regions, as demonstrated by other systematic reviews [47,51].

Therefore, operational deficiencies such as lack of water treatment have been observed despite Brazil have enhanced access to water supply networks and infrastructure since the 1940s. In 1948, shortly before implementation of the first survey included in this study, only 9% of municipalities received treated water, a deficiency even more prominent in rural areas [52]. Incomplete water treatment and deficient distribution systems are still a reality since 11.7% of Brazilian municipalities still lacked operative water treatment plants, either conventional or simplified, in 2017 [32,53].

Other conditions different from environmental factors contributed to the decrease in disease prevalence, such as the condition of household occupation, degree of urbanization, and population size. Low socioeconomic status is a known risk factor for diseases caused by parasitic infections such as schistosomiasis [27,54]. In this study, residents' housing conditions, such as acquired households and owned rather than rented dwellings, were used as a proxy for socioeconomic status. A similar conclusion was obtained in studies in Pakistan, Bangladesh, and Thailand, with families living in rented houses at increased risk of developing infectious diseases or their symptoms, including parasitic bowel diseases, compared with families who owned their housing [55–57]. Thus, the condition of home ownership was associated as a protective factor against disease, demonstrating that socioeconomic structure can produce and condition the distribution of schistosomiasis in the population.

Regarding the degree of urbanization, recent outbreaks of schistosomiasis have been prevalent in urban and peri-urban environments due to unplanned urbanization [58–62]. On the other hand, rapid urbanization has implications for infectious diseases usually described in rural areas and reduces the risk of exposure to infection in previously endemic areas [62]. Thus, the effect on epidemiological patterns of the relationship between demographic events of inter and intra-regional migratory flows with economic cycles of retraction and expansion of agricultural and industrial activities experienced by the country is undeniable. This relationship generated the model of capitalist expansion and economic growth, sometimes excluding but also enabling the last seven decades of educational, sanitary, economic, and infrastructure improvements that also resulted in changes to epidemiological patterns of infectious diseases [63,64]. Therefore, urbanization is assumed as a protective trait against the disease, which could be reverberate in the institutional feasibility of increasing and expanding public health and sanitary policies. The establishment of the Brazilian Unified Health System (SUS), including an alternative model focused on the promotion and prevention of health from the decentralization of strategies and programs for the control of schistosomiasis, is an example of these health and sanitary polices [65,66]. Other public policies, which could correspond to the changes that have occurred over the decades, are water and sanitation services, such as the National Sanitation Plan (*Plano Nacional de Saneamento–Planasa*), established in 1971 and abolished at the end of the following decade, and the current federal basic sanitation policy from Law No. 11.445/2007 and No. 14.026/2020. Although increased access to public services was considered deeply discriminatory in the 1970s regarding demographic and social criteria and currently poses risks concerning the universal access to services and human rights, they were essential instruments for expanding public water supply and sanitary sewage networks in the country [67,68].

Regarding the parasitological tests used in the surveys, although two different methods were used, the comparability between them is possible. Firstly, during the 1947–1953 IHE Brazil presented a high prevalence of schistosomiasis and a high intensity of infection, implying that the application of less sensitive diagnostic methods, such as the technique of spontaneous sedimentation in water (Hoffman, Pons et al. Janer, or HPJ technique) [69], leads to a low number of false-negative [70]. Secondly, the results obtained in the two last surveys (PECE and INPEG), which used the Kato-Katz method, could identify a greater number of true positive, with a low detection of false-negatives due to superior sensitivity of the method. It is well known that the Kato-Katz method is currently the gold standard method recommended by the WHO [3,71].

The applied statistical analysis is supported for the structure of the data and allowed revealing important results not yet studied in the country, considering the representativeness of a national sample and with historical temporality. The option of using GLMMs is based on the fact that it allowed to use the information from all municipalities of the three surveys (n = 1721) in the analysis and improving the estimation of the parameters of the model, respective standard deviations and p-values. It is well known that multilevel models (with random effects) provide better inference from grouped data (in the case of the presented study, students are grouped in municipalities which are grouped in states which are grouped in regions) since the coefficient and variance error for each explanatory variable are better estimated, avoiding the problem of underestimation of coefficients and overstatement of their significance that occur when clustering effect is not taken into account [19]. Summary statistics for the prevalence estimated by the model were consistent with those observed in the data. Sensitivity analysis shown that results obtained using the zero-inflated Poisson GLMM are consistent with those found in the restricted analysis of samples common to the three surveys.

In general, the findings of this research show that the reduction in the prevalence of schistosomiasis in Brazil over seven decades can be explained by the combination of community, demographic, socioeconomic, and specific environmental factors. The ecological design of the study, with the municipality as the unit of analysis, impairs including behavioral and other individual variables in the model, likely associated with infection. The mechanism of schistosomiasis transmission is complex and includes several conditioning factors [72]. Thus, disease control depends on preventive measures, such as early diagnosis and timely treatment, health education, surveillance and control of intermediate hosts, and basic sanitation. It is also noteworthy that the Brazilian regions differ in how their governments administer the promotion of disease control policies, especially among states that differ in aspects like location, territorial extension, and environmental and socioeconomic conditions that could interfere with the disease cycle. In line with findings of other studies, differences between forms of access and exposure to water and sanitation relate to variations in disease infection rates over time and in different regions, suggesting that the impact of access to water and sanitation is mediated by other social, behavioral, and environmental factors [73].

## Limitations

Although the results obtained in this study came from different municipalities and in different periods, consisting of non-serial temporal trend surveys, the analysis of municipalities common to the three surveys supported the other findings (see S3 Note). Other limitations must be considered when interpreting these results. The variables were collected in different periods, and such practice of collecting old census data required a process of harmonization between variables to allow comparisons. Another limitation inherent to census data includes the availability of access-restricted information on public service facilities and not the quality and availability of WaSH services. Further exploration of other data is necessary to understand the positive association between the prevalence of schistosomiasis and the availability of drinking water networks, including the effects of supply interruptions and changes in use based on water quality or behavioral and occupational habits. Finally, we cannot make conclusions on the causality of this association due to a limitation of ecological design.

To the best of our knowledge, this is the first study that used a longitudinal epidemiological design to analyze data from national prevalence surveys covering a large period of many decades. The results showed that the prevalence of schistosomiasis infection in schoolchildren in Brazilian municipalities decreased significantly over the decades. This decrease in prevalence of infection may be associated with environmental factors, urbanization, and housing conditions, which have improved over the decades. It is noteworthy that the association with water supply should be carefully interpreted and focused on other possible factors not evaluated here, confirming the need for a systemic approach. In addition, safe sanitation sewage should be widely provided to the population at the household level and other spheres of life, such as workplaces, health centers and school environments. Other national prevalence surveys and research should be conducted more continuously to monitor the disease prevalence and its determinants over time.

## Supporting information

**S1 Note. General and methodological characteristics of the three surveys National Helminthological Survey of Schoolchildren (IHE) (1947–1953), Special Schistosomiasis Control Program (PECE) (1975–1979), and National Survey of Schistosomiasis and Geohelminthiasis Prevalence (INPEG) (2011–2015).**
(DOCX)

**S2 Note. Methodological criteria related to the process of creating municipalities, used for inclusion and exclusion from the study.**
(DOCX)

**S3 Note. Supplementary statistical analysis–model definition, predictive analysis, sensitivity analysis, comparison of endemicity level distribution between the three surveys.**
(DOCX)

## Acknowledgments

We thank the task teams responsible for organizing and operationalizing the research field in all surveys. We are immensely grateful to the researcher Prof. Dr. Naftale Katz, from Instituto René Rachou/Fiocruz Minas, who assisted in making this research feasible by guiding us to the source and acquisition of data and for sharing with us his experience in conducting surveys.

## Author Contributions

**Conceptualization:** Mariana Cristina Silva Santos, Léo Heller.

**Data curation:** Mariana Cristina Silva Santos, Léo Heller.

**Formal analysis:** Mariana Cristina Silva Santos, Guilherme Lopes de Oliveira, Sueli Aparecida Mingoti, Léo Heller.

**Funding acquisition:** Léo Heller.

**Investigation:** Mariana Cristina Silva Santos, Léo Heller.

**Methodology:** Mariana Cristina Silva Santos, Guilherme Lopes de Oliveira, Sueli Aparecida Mingoti, Léo Heller.

**Project administration:** Mariana Cristina Silva Santos, Léo Heller.

**Resources:** Mariana Cristina Silva Santos, Léo Heller.

**Software:** Mariana Cristina Silva Santos, Guilherme Lopes de Oliveira, Sueli Aparecida Mingoti.

**Supervision:** Sueli Aparecida Mingoti, Léo Heller.

**Validation:** Sueli Aparecida Mingoti, Léo Heller.

**Visualization:** Mariana Cristina Silva Santos, Guilherme Lopes de Oliveira, Sueli Aparecida Mingoti, Léo Heller.

**Writing – original draft:** Mariana Cristina Silva Santos, Guilherme Lopes de Oliveira, Sueli Aparecida Mingoti, Léo Heller.

**Writing – review & editing:** Mariana Cristina Silva Santos, Guilherme Lopes de Oliveira, Sueli Aparecida Mingoti, Léo Heller.

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
