## [Decision Letter · Decision Letter 0]

16 Nov 2022

Dear Miss Santos,

Thank you very much for submitting your manuscript "Effect of environmental factors in reducing the prevalence of schistosomiasis in schoolchildren: A panel analysis of three extensive national prevalence surveys in Brazil (1950–2018)." for consideration at PLOS Neglected Tropical Diseases. As with all papers reviewed by the journal, your manuscript was reviewed by members of the editorial board and by several independent reviewers. In light of the reviews (below this email), we would like to invite the resubmission of a significantly-revised version that takes into account the reviewers' comments. 

Dear Authors,

The reviewers have raised several concerns that to my understanding could be addressed to improve the quality of the paper. Please ensure to address them all before a resubmission of a revised version. Please provide a comprehensive response to all items and comments mentioned by the reviewers and make sure to conduct a comprehensive review of the statistical analysis and language of the manuscript.

We cannot make any decision about publication until we have seen the revised manuscript and your response to the reviewers' comments. Your revised manuscript is also likely to be sent to reviewers for further evaluation.

Sincerely,

Mabel Carabali, M.D., M.Sc., Ph.D.,

Academic Editor

Cinzia Cantacessi

Section Editor

Dear Authors,

The reviewers have raised several concerns that to my understanding could be addressed to improve the quality of the paper. Please ensure to address them all before a resubmission of a revised version. Please provide a comprehensive response to all items and comments mentioned by the reviewers and make sure to conduct a comprehensive review of the statistical analysis and language of the manuscript.

Reviewer's Responses to Questions

**Key Review Criteria Required for Acceptance?**

**Methods**

-Are the objectives of the study clearly articulated with a clear testable hypothesis stated?

-Is the study design appropriate to address the stated objectives?

-Is the population clearly described and appropriate for the hypothesis being tested?

-Is the sample size sufficient to ensure adequate power to address the hypothesis being tested?

-Were correct statistical analysis used to support conclusions?

-Are there concerns about ethical or regulatory requirements being met?

Reviewer #1: Major comments: 

Study design (109): the authors suggest that three cross-sectional studies analyzed together consist of a longitudinal study. Please, explain better or review the term “longitudinal” for the study design. 

(149-151): the authors state the superiority of the Kato-Katz method, but in the first survey, spontaneous sedimentation was used. Please, explain how the authors standardized the prevalence with such different methods for Schistosomiasis diagnosis. 

Inclusion and exclusion criteria (170-193): Only 41 municipalities are repeated for the three surveys, and the authors chose 1,721 municipalities to analyze. Please, explain how it was possible to infer the prevalence fall with this data, and how to infer the independent variables, if it was selected from the first survey 907 municipalities vs 293 and 521 municipalities from the second and third surveys, respectively.

Reviewer #2: The objectives of the study are clearly articulated.

The study design does not appropriated to address the stated objectives.

Yes, the population clearly described and appropriate for the hypothesis.

Yes, the sample size is sufficient to ensure adequate power to address the hypothesis.

Not correct statistical analysis used to support conclusions.

Yes, there are concerns about ethical or regulatory requirements

Reviewer #3: The objectives of the study are clearly articulated and the study design is appropriate to address the objectives. I have no concerns regarding the ethical or regulatory requirements.

A few methodological points for the authors to consider:

1. It’s possible the information on the surveys is lacking, particularly for the second survey, but it’s unclear what the sampling strategies were for selecting municipalities, schools, and children from the population of eligible regions. Was there any random selection of municipalities or schools within them? Were all students at the selected schools examined or was there a sampling strategy applied to students? If it was all non-probabalistic, what methods were used to select/include municipalities/schools/students?

2. I believe the analysis methods used are appropriate, but the explanation is a bit confusing. It is not clear what the response is in the Poisson regression models – the authors state it is prevalence, but Poisson models generally use count as a response and then persons (or person-years) at risk as an offset for this kind of data. This is quite confusing as the authors talk about prevalence in the methods and then talk about rate ratios in results but there does not seem to be a distinction in the models fitted and the response used. For me a prevalence and rate would have a different response variable in the model. I think the outcome variable needs to be made clear in these instances and/or the language (prevalence or rate/rate ratio) needs to be tightened up. Maybe these models could be more explicitly described in an additional supplementary file.

3. There is a large amount of ‘missing’ data in the longitudinal models resulting from the changing survey sampling methods and I would like to see some consideration and discussion of this in the manuscript. Of the over 1,700 municipalities included in the longitudinal analysis, only around 400 municipalities had repeated survey data and so the sampling methodology is not necessarily consistent with conclusions that prevalence is decreasing. The authors could consider including a sensitivity analysis that includes only the municipalities with 2 or 3 surveys to see if those longitudinal patterns are similar to patterns for the full sample.

Reviewer #4: -Are the objectives of the study clearly articulated with a clear testable hypothesis stated?

The objectives of the study are clearly articulated: the goal is to examine how the prevalence of schistosomiasis has changed over time and to investigate the association between prevalence and environmental factors. 

-Is the study design appropriate to address the stated objectives?

The surveys from which the data are sourced are clearly explained. However, a major concern is that the three surveys all seem quite different. This raises the question of how appropriate it is to use these three separate datasets in a longitudinal approach (different measures are used in each survey and different sampling techniques). More on this in the specific comments by line in answer to the 'Editorial and Data Presentation Modifications' question in the review form.

-Is the population clearly described and appropriate for the hypothesis being tested?

The population is not very clearly described. In parts of the articles, the authors suggest that the results allow for inferences to be made about the impact of environmental factors on schistosomiasis in ‘Brazil’. However, in the Limitations section, the authors state that the results are only informative about this sample – this contradicts the suggestion that the results are generalisable to ‘Brazil’. 

Also, for the surveys used, it appears that different sub-populations were the target of each survey. Thus, it is difficult to understand what population the authors are attempting to understand with the sample they have. 

-Were correct statistical analysis used to support conclusions?

In relation to the statistical analysis, the article lacks a clear foundation and motivation for the techniques employed. The authors note that they use a Poisson regression. To help readers less familiar with the research on disease prevalence rates, it would be helpful to note somewhere that this is the standard approach used to examine disease prevalence. Similarly, the authors do not explain the specific implications of the ‘zero-inflated’ Poisson regression. For example, no equations are presented to help the reader understand what equation is being estimated. 

Also, in explaining the reason for selecting the final model, the authors mention using the AIC, BIC and residual variance criteria. It is unclear how comparable these criteria are for different models (Poisson vs Negative Binomial) and with different degrees of freedom. Also, do all 3 criteria consistently point to the same model? Could one criterion point to one model being most appropriate while another criterion points to another model? It would be helpful to pre-empt these questions in the discussion of the process used for model selection. 

The discussion of the GLMMs lacked clarity. There is a lot of discussion in this section (lines 243-257) about modelling of ‘data’, without any specific discussion of the relevant variables. E.g., in the discussion of random effects, it is unclear what variable this specifically refers to. It would be useful to clarify. 

It is a little concerning that so many of the key explanatory variables are measured using projections. I appreciate the authors transparency on this. Perhaps if they presented a table or outlined in the writing how many observations are affected, that would help to shed light on the likely implications of this. 

More comments on the statistical methods can be found in the specific comments by line in answer to the 'Editorial and Data Presentation Modifications' question in the review form.

-Is the sample size sufficient to ensure adequate power to address the hypothesis being tested?

-Are there concerns about ethical or regulatory requirements being met?

The sample size is sufficient and there are no ethical concerns.

**Results**

-Does the analysis presented match the analysis plan?

-Are the results clearly and completely presented?

-Are the figures (Tables, Images) of sufficient quality for clarity?

Reviewer #1: Please, clarify the affirmation of the findings of this research in lines 488-490.

Reviewer #2: Some of the analysis presented match with the analysis plan.

The results are not clearly and completely presented.

The figures (Tables, Images) are not of sufficient quality for clarity

Reviewer #3: The results (including tables and figures) are clearly presented and of sufficient quality. Please see my previous comment in the methods section regarding the description of the analysis methods.

Reviewer #4: -Does the analysis presented match the analysis plan?

The authors present the results form one Poission model (with zero inflation). Earlier in the article, the authors make it clear to the reader that this is the model that has been chosen as the most appropriate based on a variable exclusion process and a comparison of the AIC, BIC etc. criteria. 

-Are the results clearly and completely presented?

The results are quite clearly presented. However, I felt the authors could have done a little more to interpret the relevant coefficient and RRs. For example, on line 384, the authors note this model predicts that an increase in coverage of the sewage system from 20% to 100% is associate with a 27.4% reduction in the average prevalence. It would be informative to explain how the 27.4% figure is reached. 

The results are somewhat incomplete as only one model is examined. It would be interesting to examine one or two more specifications. It would also be helpful to see how sensitive the results are to the model specification. More on this in the specific comments by line in answer to the 'Editorial and Data Presentation Modifications' question in the review form

-Are the figures (Tables, Images) of sufficient quality for clarity?

For the most part, the tables/figures are very clear with excellent notes to explain acronyms and keys. Some minor problems are noted in answer to the 'Editorial and Data Presentation Modifications' question in the review form.

**Conclusions**

-Are the conclusions supported by the data presented?

-Are the limitations of analysis clearly described?

-Do the authors discuss how these data can be helpful to advance our understanding of the topic under study?

-Is public health relevance addressed?

Reviewer #1: -Are the conclusions supported by the data presented? No. 

-Are the limitations of analysis clearly described? Yes.

-Do the authors discuss how these data can be helpful to advance our understanding of the topic under study? Yes

-Is public health relevance addressed? Yes.

Limitations:

The method, as described above, is the main limitation of the current paper. The authors should carefully review and better explain how to solve the bias.

Reviewer #2: Some part of the conclusions are supported by the data.

The limitations of the analysis are partially described.

Yes, the authors discuss how these data can be helpful to advance our understanding of the topic under study.

Yes, public health relevance is addressed.

Reviewer #3: The discussion of the findings is detailed and thoroughly linked to other relevant literature. The authors have discussed how the findings can be used to improve public health outcomes for children. I'd like to see a sensitivity analysis (see previous comment) to help support the conclusions and the limitation of missing data should also be discussed.

Reviewer #4: -Are the conclusions supported by the data presented?

Yes, for the most part, the conclusions relate to the data analysis. The discussion section could be written using clearer language to help communicate the findings (more on this in the specific comments by line in answer to the 'Editorial and Data Presentation Modifications' question in the review form). However, the authors make recommendations about offering education on sanitary sewage and health: it is unclear how this recommendation emerges from this study. 

-Are the limitations of analysis clearly described?

I felt the first point in the Limitations section, about the results only being valid for the observed sample and not generalisable, seems to contradict the suggestion earlier on that these results shed light on schistosomiasis prevalence in Brazil more generally (more on this in the specific comments by line in answer to the 'Editorial and Data Presentation Modifications' question in the review form).

-Do the authors discuss how these data can be helpful to advance our understanding of the topic under study?

The authors provide a very interesting discussion about the (at first glance) perplexing result that increased access to piped water has a negative association with prevalence. The authors highlight that this result shows that access to piped water and the impact on health is more nuanced than simply a binary question of whether a household has access or not. This discussion highlights that access to piped water must be accompanied by a reliable source to ensure its potential positive impact is realised. The authors also discuss other relevant environment factors that aid our understanding of the topic. More on this in the specific comments by line in answer to the 'Editorial and Data Presentation Modifications' question in the review form.

-Is public health relevance addressed?

The relevance to public health is clear throughout this article. The topic of schistosomiasis is a listed interest for the PLOS Neglected Tropical Diseases journal. Therefore, this topic is highly relevant to this journal. The relationship between environmental factors and the prevalence of any NTD is important for understanding how to tackle the disease and reduce the prevalence.

**Editorial and Data Presentation Modifications?**

Reviewer #1: Minor comments:

Table 1: Municipal GDP per capita - please correct PECE period “975”.

We suggest expanding the introduction section with some elements of the discussion, like the factors related to the availability of piped water (422, 434) or the increased risk related to rented houses “compared with families who owned their housing” (456-459). 

We could not find the reference for Grimes et al (2014) - line 500. 

We suggest reviewing line 505. If this study is not generalizable, then how could affirm that this study indicates a decrease in the prevalence of schistosomiasis over seven decades from the analyzed municipalities? 

Supplementary Materials:

none to declare. 

The tables and pictures are adequate and well explored. 

Title and abstract:

Title: none to declare;

abstract: please review the recommendations and, if possible, some of the suggestions. 

We indicate a “major review” of this article before acceptance.

Reviewer #2: Accepted

Reviewer #3: The abstract mentions the prevalence of schistosomiasis was mediated by environmental factors – the analysis method was not a mediation analysis, so I’d suggest the authors revise their phrasing here to avoid confusion.

In Table 1, the description of the literacy rate doesn’t include how literacy was defined in the numerator and currently only states “Number of people aged 15 years old or Older”

Line 199 should say explanatory variable not exploratory.

Line 244 should say generalized linear mixed models not generalized mixed linear models to align with the acronym GLMM.

Table 2 and associated text - the term ‘range’ is usually used for the difference between maximum and minimal values.

Line 350 "the study proves" is very strong language and should perhaps be modified to something along the lines of 'the study has found significant associations between environmental factors and schistosomiasis'.

Reviewer #4: In this section, I have provided comments for the authors relating to specific lines in the manuscript: 

Line 10: grammatical issue: change ‘was examined’ to ‘were examined’

Line 12: Unclear on the meaning of ‘were adjusted’. I expect a phrase like ‘were estimated’ here. 

Line 22: Unclear on the meaning of ‘other ways of contact with watercourses’. This phrase is a little clunky. 

Line 38: ‘momentum historical’ doesn’t make sense.

Line 46: The part of the sentence that comes after the comma doesn’t seem to fit with the first part of the sentence. You appear to say that prevalence of schistosomiasis decreased, but that this suggests that better approaches are needed for controlling the disease.

Line 71: Wording issue: ‘control measures conditions’. Unclear what you are trying to say in this phrase. 

Lines 84-100: In this section, you examine how the prevalence rate has changed over time. However, I am concerned as to whether this comparison is valid as it appears that each survey targeted a different Brazilian sub-population.

Line 110: The wording ‘a longitudinal study on three panels’ sounds a little odd. Perhaps it would be clearer to say something like: a longitudinal study covering three waves of data?

Line 126: The word ‘besides’ is unclear in this context.

Line 144: By stating that there were 197,564 examinations of schoolchildren, it may suggest to the reader that the unit of observation is at the child level (whereas it is at the municipality level). Perhaps it would be more helpful to focus on whether a representative sample of children were examined in each municipality as well as comment on the numbers involved. 

Line 146-160: In this section, the authors describe differences in the techniques used in the different surveys: different methods to measure prevalence and even different age ranges. This again raises concern about how appropriate it is to compare prevalence rates over time and how appropriate it is to combine these waves of data into one longitudinal dataset. Looking at the results section, I can see that you do appear to control for this with wave dummy variables in your model? It would be good to address this more clearly in the writing. 

Line 161: Figure 1 is very clear for the most part. There are a coupe of places where it looks like the wording has not been translated into English: ‘Alunos Examinados’?

Line 169, Table 1: The description of the prevalence of schistosomiasis measure raises some concern. The numerator used to calculate the prevalence is the number of students with a positive stool sample. However, earlier the authors suggested that different approaches to the stool testing have been used in different surveys. E.g., the Kato-Katz method was not used in all surveys. The denominator used to calculate the prevalence also raises some concerns as it is unclear what population is represented by the sample of 'all examined students'. This comment echoes concerns in relation to earlier parts of the article on how appropriate it is to compare and combine all waves of data.

Line 169, Table 1: I believe the literacy rate description is missing some words. You need to specify that the numerator includes only people age 15 or older 'who are literate'? 

Line 204-205: Unclear what is meant by the phrase ‘seeking to adjust the uniformity and consistency over time’. 

Line 213 – 229: It is a little concerning that so many of the key explanatory variables are constructed using projections. I appreciate the authors transparency on this. Perhaps if they presented a table or outlined in the writing how many observations are affected, that would help to shed light on the likely implications of this. 

Line 231: Unclear what is meant by the statement ‘the national surveys are not continuous’. 

Lines 243-257: The discussion of the GLMMs lacked clarity. There is a lot of discussion in this section about modelling of ‘data’, without any specific discussion of the relevant variables. E.g., in the discussion of random effects, it is unclear what variable this specifically refers to. It would be useful to clarify. 

Lines 259-261: In explaining the reason for selecting the final model, the authors mention using the AIC, BIC and residual variance criteria. It is unclear how comparable these criteria are for different models (Poisson vs Negative Binomial) and with different degrees of freedom. Also, do all 3 criteria consistently point to the same model? Could one criterion point to one model being most appropriate while another criterion points to another model? It would be helpful to pre-empt these questions in the discussion of the process used for model selection. 

Line 274, Table 2: In the notes about the GDP measure, I expected it to state the base year for the GDP measure (as Table 1 stated that GDP is measures in constant R$). 

Line 301-302: It would be helpful to offer a little more clarity on how many models were considered. That could be discussed here, or earlier in the article. 

Lines 301-305: I think it would help the reader to understand the coefficients and RRs presented in the model if the equation of the model you are estimating was explicitly stated. 

Line 308. Table 4: I wonder should the estimate for ln(population) read as 0.138 rather than 0.148? In the writeup you state than the associated effect is a 13.8% decrease? If the explanatory variable is in logs, I believe this effect can be interpreted as an elasticity? 

Line 323: Typo – change 32.5% to 35.2%

Lines 355-362: I found the writing in this section to be unclear.

Line 384: It is unclear how the 27.4% has been calculated.

Line 415-418: It is unclear what point is being made in the sentence beginning ‘An intervention study…’ 

Line 489: The authors state that the findings from this research show ‘that the reduction in the prevalence of schistosomiasis in Brazil over seven decades can be explained by ….’ But later in line 505, the authors state the results cannot be generalized outside of the specific sample. Later on line 519, the authors also state that the prevalence decreased significantly over the decades in Brazilian municipalities. There is a lack of clarity on whether these results should be used to make inference about the entire Brazilian population or if the results cannot be generalised outside of the sample. 

Line 524 – 526: The authors make recommendations about offering education on sanitary sewage and health. It is unclear how this recommendation emerges from this study.

**Summary and General Comments**

Reviewer #1: Summary of strengths, weaknesses, and overall contribution:

This is an ecological study that analyzes three different Brazilian Schistosomiasis surveys and proposes a series of factors that could explain a reduction in the prevalence of this disease among schoolchildren between 1950-2018.

The authors design a robust statistical inference framework and declare that, although there are many differences in the surveys (methods for diagnosis and sampling, for example), “the findings of this study indicate a decrease in the prevalence of schistosomiasis over seven decades in schoolchildren from the analyzed Brazilian municipalities, mediated by environmental factors and social conditions.”

This paper could contribute to the implementation of public policies, like larger access for sanitization, planned urbanization, and a proposition for a long-term policy for household ownership. 

The main problem of this paper is methodological: the bias for the lack of “harmonization between variables to allow comparisons”; the fact that only 41 municipalities repeated the three surveys together; the diagnosis method is different in the first survey from the other ones; the second survey “consisted of a non-probabilistic sample”. This problem compromises all the findings of this paper.

Reviewer #2: The language should be checked, and some parts of the paper may require additional work or clarification.

The paper can .be published upon addressing the attached comments.

Reviewer #3: The manuscript is well-written, easy to follow, and the authors have been transparent about the limitations of the datasets used and how they have dealt with these in their analysis.

Reviewer #4: Dear Authors

Thank you for allowing me to review this manuscript entitled “Effect of environmental factors in reducing the prevalence of schistosomiasis in schoolchildren: A panel analysis of three extensive national prevalence surveys in Brazil (1950–2018)”. 

I sincerely apologise for my delayed response. The last month was very busy for me. It was my busiest teaching period and my husband, my infant daughter, and I were all sick in quick succession (nothing serious, thankfully, but time-consuming). 

I have now reviewed the manuscript. I believe that this article makes a valuable contribute to the Neglected Tropical Diseases (NTD) literature. The main strengths of the article can be summarised as follows:

1. The article explored a rich dataset collected across multiple decades. The authors provide a thorough discussion of how data were collected in these surveys. This allows for a very interesting examination of how schistosomiasis prevalence has changed over time. 

2. It is very informative to examine associations with environment factors and particularly interesting to carry this out at the Brazilian municipality level. 

3. The authors provide a very interesting discussion about the (at first glance) perplexing result that increased access to piped water has a negative association with prevalence. The authors highlight that this result shows that access to piped water and the impact on health is more nuanced than simply a binary question of whether a household has access or not. This discussion highlights that access to piped water must be accompanied by a reliable source to ensure its potential positive impact is realised. 

4. The authors indicate that a careful process was used to choose the most appropriate statistical model for the data. 

5. The authors are clear and transparent about the inclusion and exclusion criteria used to determine the final estimation sample.

6. The relevance to public health is clear throughout this article. The topic of schistosomiasis is a listed interest for the PLOS Neglected Tropical Diseases journal. Therefore, this topic is highly relevant to this journal. The relationship between environmental factors and the prevalence of any NTD is important for understanding how to tackle this problem and reduce the prevalence. 

However, there are some major issues that I feel need to be addressed:

1. A clearer definition of the population you are examining needs to be given. I have some concerns about the combination of waves of survey data where the target population appears to differ for each survey. It is unclear what inferences can be made from your results. As noted in my comments by line (in answer to the 'Editorial and Data Presentation Modifications?' question in the review from), there are some contradictions in your writing. In some parts, you appear to suggest that the results allow for a better understanding of the schistosomiasis prevalence rate in all of Brazil; in other parts, you suggest that the results cannot be generalised outside of the specific sample you are examining.

2. The article lacks a clear foundation and motivation for the statistical techniques employed. For example, no equations are presented to help the reader understand what equation is being estimated.

3. I believe the interpretation of the model results could be more clearly explored. Also, it would be beneficial to carry out some sensitivity analysis. 

4. There are many minor wording issues throughout the article which, when combined, sum up to constitute a major issue. The issues are listed by line in answer to the ‘Editorial and Data Presentation Modifications’ question in the review form. 

I hope that the comments provided in this review will help you to revise and improve this article. Thank you again for producing a highly topical study that has the potential to improve the public health guidance for NTDs.

PLOS authors have the option to publish the peer review history of their article (what does this mean?). If published, this will include your full peer review and any attached files.

Reviewer #1: Yes: Thiago Figueiredo de Castro

Reviewer #2: No

Reviewer #3: No

Reviewer #4: No

Figure Files:

Data Requirements:

Please note that, as a condition of publication, PLOS' data policy requires that you make available all data used to draw the conclusions outlined in your manuscript. Data must be deposited in an appropriate repository, included within the body of the manuscript, or uploaded as supporting information. This includes all numerical values that were used to generate graphs, histograms etc.. For an example see here: http://www.plosbiology.org/article/info:doi%2F10.1371%2Fjournal.pbio.1001908#s5.
---

## [Decision Letter · Decision Letter 1]

7 Mar 2023

Dear Miss Santos,

Thank you very much for submitting your manuscript "Effect of environmental factors in reducing the prevalence of schistosomiasis in schoolchildren: A panel analysis of three extensive national prevalence surveys in Brazil (1950–2018)." for consideration at PLOS Neglected Tropical Diseases. As with all papers reviewed by the journal, your manuscript was reviewed by members of the editorial board and by several independent reviewers. In light of the reviews (below this email), we would like to invite the resubmission of a significantly-revised version that takes into account the reviewers' comments. 

Thank you for submitting a revised version of your manuscript.

However, the reviewers have raised some remaining concerns about the manuscript.

Please address and provide the respective considerations from the reviewers carefully.

We cannot make any decision about publication until we have seen the revised manuscript and your response to the reviewers' comments. Your revised manuscript is also likely to be sent to reviewers for further evaluation.

Sincerely,

Mabel Carabali, M.D., M.Sc., Ph.D.,

Academic Editor

Cinzia Cantacessi

Section Editor

Thank you for submitting a revised version of your manuscript.

However, the reviewers have raised some remaining concerns about the manuscript.

Please address and provide the respective considerations from the reviewers carefully.

Reviewer's Responses to Questions

**Key Review Criteria Required for Acceptance?**

**Methods**

-Are the objectives of the study clearly articulated with a clear testable hypothesis stated?

-Is the study design appropriate to address the stated objectives?

-Is the population clearly described and appropriate for the hypothesis being tested?

-Is the sample size sufficient to ensure adequate power to address the hypothesis being tested?

-Were correct statistical analysis used to support conclusions?

-Are there concerns about ethical or regulatory requirements being met?

Reviewer #1: Major comments: 

none do declare. We appreciate that all the four main points were appropriately answered. 

Minor comments:

The use of P-value: Pearson's Chi-squared statistical test comparing the period with 2013 in S3 - supplementary material is fundamental for validation of this research. Please, clarify this state in the S3 - supplementary material: “As shown in Table S3.3, the p-values of the tests indicated that the distributions were similar to the third survey (p-value=0.598 for 1950; p-value=0.183; for 1977; p-value=0.145, when considering common municipalities in all three surveys).Considering these results, it can be concluded that, although there are differences in the form of data collection regarding the selection of municipalities, the three samples are comparable in terms of the endemicity degree criterion used in the 2011-2015 survey.”

Reviewer #2: Yes

Reviewer #4: Thank you to the authors for addressing my comments on this section. 

My remaining concerns:

I think the key aspects of the Data Analysis are lost, at times, with convoluted language. I, personally, would find it helpful to see an equation/equations to better explain the methodology (one equation showing the main estimation framework - no need for equations for the sensitivity analysis, model selection process). However, I understand that may not be everyone's preference. I think it would be helpful to move a lot of the more convoluted explanations relating to the choice of model to the supplementary material. This would allow readers to focus on your final chosen model and how to interpret it. 

Similarly, I think that many of the key points about the data are lost due to overwhelming detail on the data selection process etc. To begin with, I think it is unnecessary to provide a rough summary of the data in the introduction before the more detailed discussion in the Methods section which follows. Perhaps refer to it much more briefly in the introduction and explain that the more detailed discussion will follow. I also think it would be helpful to reduce the discussion of the data selection process somewhat in the main text and keep that finer detail for the supplementary material.

**Results**

-Does the analysis presented match the analysis plan?

-Are the results clearly and completely presented?

-Are the figures (Tables, Images) of sufficient quality for clarity?

Reviewer #1: Major comments: 

none do declare. We appreciate that all the four main points and some of the minor comments were appropriately answered. 

Limitations:

The method, as described above, is the main limitation of the current paper. The authors should carefully review and better explain how to solve the bias. 

Supplementary Materials:

none to declare. 

The tables and pictures are adequate and well explored. 

Title and abstract:

Title: none to declare;

abstract: please review the recommendations and, if possible, some of the suggestions.

Reviewer #2: The analysis presented not match with the analysis plan.

Tables are not clear.

Reviewer #4: Thank you to the authors for addressing my comments on this section. I think the Results section is very clear for the most part. Some more specific comments on this section can be found in my line-by-line comments below.

**Conclusions**

-Are the conclusions supported by the data presented?

-Are the limitations of analysis clearly described?

-Do the authors discuss how these data can be helpful to advance our understanding of the topic under study?

-Is public health relevance addressed?

Reviewer #1: none to declare

Reviewer #2: The limitations of analysis are not clearly described

Reviewer #4: Thank you to the authors for addressing my comments on this section. I think the Conclusions section is very clear for the most part. Some more specific comments on this section can be found in my line-by-line comments below.

**Editorial and Data Presentation Modifications?**

Reviewer #1: We noted that it is necessary to carefully review some editorial issues (like the lines 18, 62, 88) before submitting the final version of the article.

Reviewer #2: Minor Revision

Reviewer #4: Overall, one major concern is that the writing is quite unclear in places. There are too many wording issues for me to specifically note all of them. I have noted some of these issues below. Essentially, I believe this article would benefit from re-reading and editing. 

Thank you to the authors for addressing my previous line-by-line comments. I think it may be helpful for me to provide my line-by-line comments relating to the latest manuscript too. See below:

Line 18: 'inverse association with the water supply' is unclear. You need to indicate whether you mean piped water supply etc to help the author understand if this 'inverse association' represents a favourable or unfavourable association. 

Line 82: Meaning of 'exclusive' is unclear here. Perhaps you are trying to get across that it can not be treated as a silo. 

Lines 84-90: You do not indicate immediately here that this survey covered prevalence of schistosomiasis. On line 90 you then say 'the disease' but it not clear what disease you are referring to. 

Line 95-106: I feel this section raises lots of questions for the reader because it skims over the data. I wonder could this part be removed/reduced and save the data discussion for the more detailed section that comes next. 

Line 110: I think the word 'periods' or 'waves' would be more appropriate than 'panels'. A panel implies a longitudinal dataset that already consists of multiple periods. If this is what you are trying to communicate, it is fine to leave the writing as it is. My understanding, however, is that each of the three datasets are treated as one midpoint in time.

Line 112: Saying that the 'explanatory variables' represent population groups does not make sense to me. This would only make sense if you were creating subgroups based on the explanatory variables. But I don't think that is what you are trying to communicate. 

Line 146-147: It doesn't read well to say that 'municipalities' were 'conducted'.

Lines 180-190: Could this section possibly be incorporated in the supplementary material?

Lines 207-209: Could this section be moved to supplementary note 2 where further information is available relating to data projections in the table of supplementary note 2. Currently, the table in supplementary note 2 is difficult to understand as it on its own with no further narrative. 

Line 234: I think the word 'waves' or 'periods' would be better than 'panels'

Table 2: The GDP per capita units are not clear. Is it measured in thousands or tens of thousands Brazilian reais?

Table 3: Northeast subtotal % formatting issue. Dash rather than a period for the decimal place. 

Table 3: Make sure you consistently use a decimal point symbol. For the Southeast subtotal % 2010-2015 you use a comma instead of a period symbol. Perhaps double check to ensure formatting consistency. 

Lines 328-350: Much of this is repeating the information that is in the supplementary section. I believe it would be better to focus on the model you are using and how to interpret it. You could also briefly mention that criteria were used for choosing this model based on its goodness of fit but save the detail on model selection and sensitivity for the supplementary sections. 

Line 364: It would be helpful for the reader to consider what an increase of 1 in a logged explanatory variable means. Would it be more helpful to interpret the population coefficient by stating how this relates to percentage changes when measured in logs? 

Line 372: You state that 'based on these results' the decreasing prevalence between 1977 and 2013 was estimated. It is unclear how you determined 84.2% based on these results. Could you clarify?

Lines 384-394: It would be helpful to explain more clearly why the model estimates an increases in prevalence between 1950-1977 while the descriptive statistics indicate a decrease. Why does prevalence increase (between these two points) when the other variables in the model are held constant? You state that this could be the result of the sampling procedures used which suggests that this may be confounding your results. This then raises the question as to whether the sampling procedure was sound to begin with. 

Line 422: 'immediate host for fetching water' is unclear. 

Line 538: Am I correct in thinking that the use of random effects allows for better precision in this context but does not lead to better estimation of the coefficient itself. The writing here implies that there is somehow better estimation of the coefficient itself too?

Line 572: I do not think it's correct to call this an 'international' study when it is focused on Brazil. 

Supplementary Note S1: You refer to a power analysis. I believe you are explaining the power analysis conducted by previous authors rather than a power analysis carried out by you? Could you include a reference to the previous authors in this paragraph if this is the case? 

Supplementary Note S2, at the beginning of page 2: you mention that there were no significant differences between groups. It is not clear what variable is being tested. 

Supplementary Note S2, Table on page 2: For consistency with the rest of the manuscript, use the period symbol for decimal places.

Supplementary Note S2, Table on page 2: I think this table needs more explanation. For the first period you state the n=5442. But for the two variables with projections, there are 907 observations. So it seems like the 5442 is 907*6. However, this is a lot of work for the reader! Given that so many of the key explanatory variables are constructed using projections, I think the projection approach needs more careful discussion than is currently in the manuscript.

**Summary and General Comments**

Reviewer #1: Summary of strengths, weaknesses, and overall contribution:

This is an ecological study that analyzes three different Brazilian Schistosomiasis surveys and proposes a series of factors that could explain a reduction in the prevalence of this disease among schoolchildren between 1950-2018.

The authors design a robust statistical inference framework and declare that, although there are many differences in the surveys (methods for diagnosis and sampling, for example), “the findings of this study indicate a decrease in the prevalence of schistosomiasis over seven decades in schoolchildren from the analyzed Brazilian municipalities, mediated by environmental factors and social conditions.”

This paper could contribute to the implementation of public policies, like larger access for sanitization, planned urbanization, and a proposition for a long-term policy for household ownership. 

The methodological issues were corrected in this article reviewed version.

Reviewer #2: Thank you for sending me back the article.

From my perspective, I raised some comments/suggestions, which are not all addressed. The different observations are listed below.

1-A map of Brazil showing the administrative divisions (regions) and/or a description of how the administrative regions differ from states (level 2) and municipalities (if possible) would be helpful in explaining some of the differences in prevalence (as you mentioned in the methodology that the municipality as the unit of analysis).

2-A backward selection procedure was used to identify the significant fixed effects, considering a 25% significance level for the removal of an explanatory variable. Why the authors considered 25%?

3-The results of the Negative Binomial regression model for the various periods are not shown in the manuscript.

4-The criteria values (residual variance, AIC, and BIC) generated by the different models should be presented in the manuscript.

5-Line 312, the value described is not correct [758 (44.0%)]. It should be 788 (45.78%).

6-The percentage of the subtotal in table 3 for the Northeast region from 1947 to 1952 is incorrect, also the number of municipalities from 1975 to 1979 (114 vs 144).

Reviewer #4: Dear Authors

Thank you for allowing me to review the latest version of your manuscript entitled “Effect of environmental factors in reducing the prevalence of schistosomiasis in schoolchildren: A panel analysis of three extensive national prevalence surveys in Brazil (1950–2018)”.

I sincerely apologise for my delayed response. This year has, unfortunately, been extremely busy. 

I have now reviewed the latest manuscript. I believe that this article makes a valuable contribute to the Neglected Tropical Diseases (NTD) literature. As noted previously, there are multiple strengths to this article. I previously summarised these strengths as follows:

1. The article explored a rich dataset collected across multiple decades. The authors provide a thorough discussion of how data were collected in these surveys. This allows for a very interesting examination of how schistosomiasis prevalence has changed over time.

2. It is very informative to examine associations with environment factors and particularly interesting to carry this out at the Brazilian municipality level.

3. The authors provide a very interesting discussion about the (at first glance) perplexing result that increased access to piped water has a negative association with prevalence. The authors highlight that this result shows that access to piped water and the impact on health is more nuanced than simply a binary question of whether a household has access or not. This discussion highlights that access to piped water must be accompanied by a reliable source to ensure its potential positive impact is realised.

4. The authors indicate that a careful process was used to choose the most appropriate statistical model for the data.

5. The authors are clear and transparent about the inclusion and exclusion criteria used to determine the final estimation sample.

6. The relevance to public health is clear throughout this article. The topic of schistosomiasis is a listed interest for the PLOS Neglected Tropical Diseases journal. Therefore, this topic is highly relevant to this journal. The relationship between environmental factors and the prevalence of any NTD is important for understanding how to tackle this problem and reduce the prevalence.

However, there are some remaining issues as noted in response to the Review Questions above. In particular, I think your wording is quite unclear at times and not at the standard required for PLOS. Therefore, I feel the manuscript would benefit from further editing. 

I hope that the comments provided in this review will help you to revise and improve this article. Thank you again for producing a highly topical study that has the potential to improve the public health guidance for NTDs.

PLOS authors have the option to publish the peer review history of their article (what does this mean?). If published, this will include your full peer review and any attached files.

Reviewer #1: Yes: Thiago Figueiredo de Castro

Reviewer #2: No

Reviewer #4: No

Figure Files:

Data Requirements:

Please note that, as a condition of publication, PLOS' data policy requires that you make available all data used to draw the conclusions outlined in your manuscript. Data must be deposited in an appropriate repository, included within the body of the manuscript, or uploaded as supporting information. This includes all numerical values that were used to generate graphs, histograms etc.. For an example see here: http://www.plosbiology.org/article/info:doi%2F10.1371%2Fjournal.pbio.1001908#s5.
---

## [Decision Letter · Decision Letter 2]

3 Jun 2023

Dear Miss Santos,

We are pleased to inform you that your manuscript 'Effect of environmental factors in reducing the prevalence of schistosomiasis in schoolchildren: an analysis of three extensive national prevalence surveys in Brazil (1950–2018).' has been provisionally accepted for publication in PLOS Neglected Tropical Diseases.

Best regards,

Mabel Carabali, M.D., M.Sc., Ph.D.,

Academic Editor

Cinzia Cantacessi

Section Editor

Thank you for taking into consideration all reviewer's comments.

Reviewer's Responses to Questions

**Key Review Criteria Required for Acceptance?**

**Methods**

-Are the objectives of the study clearly articulated with a clear testable hypothesis stated?

-Is the study design appropriate to address the stated objectives?

-Is the population clearly described and appropriate for the hypothesis being tested?

-Is the sample size sufficient to ensure adequate power to address the hypothesis being tested?

-Were correct statistical analysis used to support conclusions?

-Are there concerns about ethical or regulatory requirements being met?

Reviewer #1: We appreciate that all suggestions were considered. None else to declare.

Reviewer #2: The objectives, study methodology, and sample size used for analysis meet the criteria. Furthermore, correct statistical analysis was used and there are no ethical concerns.

Reviewer #4: (No Response)

**Results**

-Does the analysis presented match the analysis plan?

-Are the results clearly and completely presented?

-Are the figures (Tables, Images) of sufficient quality for clarity?

Reviewer #1: We appreciate that all suggestions were considered. None else to declare.

Reviewer #2: The analysis provided corresponds to the analysis plans, and the results are straightforward and comprehensively presented. Furthermore, the figures (Tables, Images) are clear (of high quality).

Reviewer #4: (No Response)

**Conclusions**

-Are the conclusions supported by the data presented?

-Are the limitations of analysis clearly described?

-Do the authors discuss how these data can be helpful to advance our understanding of the topic under study?

-Is public health relevance addressed?

Reviewer #1: We appreciate that all suggestions were considered. None else to declare.

Reviewer #2: The conclusion meets all of the requirements.

Reviewer #4: (No Response)

**Editorial and Data Presentation Modifications?**

Reviewer #1: We appreciate that all suggestions were considered. None else to declare.

We recommend "Accept" this last version for publication.

Reviewer #2: Accept

Reviewer #4: (No Response)

**Summary and General Comments**

Reviewer #1: This is an ecological study that analyzes three different Brazilian Schistosomiasis surveys and proposes a series of factors that could explain a reduction in the prevalence of this disease among schoolchildren between 1950-2018.

The authors design a robust statistical inference framework and declare that, although there are many differences in the surveys (methods for diagnosis and sampling, for example), “the findings of this study indicate a decrease in the prevalence of schistosomiasis over seven decades in schoolchildren from the analyzed Brazilian municipalities, mediated by environmental factors and social conditions.”

This paper could contribute to the implementation of public policies, like larger access for sanitization, planned urbanization, and a proposition for a long-term policy for household ownership.

The methodological issues were corrected in this article reviewed version, and the minor comments were considered. The scope is clear and fits the purpose of the PNTD.

Reviewer #2: Thank you again for sending me the article.

From my side, all the comments/suggestions are addressed.

After the editor has formatted the paper slightly, it can be published.

Reviewer #4: Thank you to the authors for carefully addressing all of my comments. I am happy that the authors have incorporated changes based on these comments and I believe the article can now be 'accepted'.

A final note: A few of the inserted changed (based on my comments) have occasional minor typos, so please re-read for a final time to catch any of these minor typos. I trust the authors to do the final read and catch any final wording/grammatical issues. Therefore, I do not need to review any further versions.

Thank you to the authors for their hard work and congratulations on this valuable piece of research.

PLOS authors have the option to publish the peer review history of their article (what does this mean?). If published, this will include your full peer review and any attached files.

Reviewer #1: No

Reviewer #2: **Yes: **Dr. Thierno Souleymane Barry

Reviewer #4: No

---

## [Editor Report · Acceptance letter]

12 Jul 2023

Dear Miss Santos,

We are delighted to inform you that your manuscript, "Effect of environmental factors in reducing the prevalence of schistosomiasis in schoolchildren: an analysis of three extensive national prevalence surveys in Brazil (1950–2018).," has been formally accepted for publication in PLOS Neglected Tropical Diseases.

Best regards,

Shaden Kamhawi

co-Editor-in-Chief

Paul Brindley

co-Editor-in-Chief
